# High Precision Optical Tracking System Based on near Infrared Trinocular Stereo Vision

**DOI:** 10.3390/s21072528

**Published:** 2021-04-04

**Authors:** Songlin Bi, Yonggang Gu, Jiaqi Zou, Lianpo Wang, Chao Zhai, Ming Gong

**Affiliations:** 1Department of Precision Machinery and Precision Instrumentation, University of Science and Technology of China, Hefei 230027, China; bisl001@mail.ustc.edu.cn (S.B.); zjq007@mail.ustc.edu.cn (J.Z.); wlp0219@mail.ustc.edu.cn (L.W.); 2Experiment Center of Engineering and Material Science, University of Science and Technology of China, Hefei 230027, China; zhaichao@ustc.edu.cn (C.Z.); gongming@ustc.edu.cn (M.G.)

**Keywords:** optical tracking system, trinocular stereo vision, dynamic tracking

## Abstract

A high precision optical tracking system (OTS) based on near infrared (NIR) trinocular stereo vision (TSV) is presented in this paper. Compared with the traditional OTS on the basis of binocular stereo vision (BSV), hardware and software are improved. In the hardware aspect, a NIR TSV platform is built, and a new active tool is designed. Imaging markers of the tool are uniform and complete with large measurement angle (>60°). In the software aspect, the deployment of extra camera brings high computational complexity. To reduce the computational burden, a fast nearest neighbor feature point extraction algorithm (FNNF) is proposed. The proposed method increases the speed of feature points extraction by hundreds of times over the traditional pixel-by-pixel searching method. The modified NIR multi-camera calibration method and 3D reconstruction algorithm further improve the tracking accuracy. Experimental results show that the calibration accuracy of the NIR camera can reach 0.02%, positioning accuracy of markers can reach 0.0240 mm, and dynamic tracking accuracy can reach 0.0938 mm. OTS can be adopted in high-precision dynamic tracking.

## 1. Introduction

Optical tracking system (OTS), widely used in medical navigation and robots, can accurately estimate positions and orientations of surgical tools in real-time [1,2]. The doctor’s eyes can be replaced by OTS to measure key areas in some surgeries. Owing to high positioning precision, surgical trauma can be minimized. A typical OTS mainly consists of binocular stereo vision (BSV) and optical tools. The tools have a series of optical markers. The calibrated BSV captures surgical tools from different angles, and positions of markers can be computed by measuring their distances to cameras through triangulation. Distances between markers and working point of the tool are known, thus, coordinates of the working point can be computed by using geometric constraint. This method can be used in many fields, where the working point is blocked or difficult to be measured. A large number of researchers have made their own contributions to the research in the past decades.

The early OTS is mainly used in the industrial field [3]. Since the 1990s, with the development of computer aided technology and optical positioning technology, OTS was gradually applied in the medical field and robotics [4]. Kuos et al. [5] integrated OTS into robots and developed a set of surgical marker emitters for surgical navigation applications. Lee et al. [6] proposed a navigation system combining multiple space-guiding trackers for cerebral endovascular surgery. This system not only provided 2D and 3D visualization of medical images but also integrated an ICP-based algorithm to increase the registration accuracy. Lin et al. [7] developed an accurate and low-cost near infrared (NIR) OTS for the real-time tracking of optical tools. Mean square errors of distances between light-emitting points on optical tools were less than 0.3 mm. Wang et al. [8] combined the optical scanning, visual tracking, and integrated mobile robot to provide accurate noncontact 3D measurements of large-scale complex components, thus, the application of OTS is further extended.

Most existing research studies of OTS are extensions of the application field, and the constructions of OTS remain mostly in BSV. OTS in the market is occupied by BSV. For example, the Northern Digital Inc (NDI) Polaris OTS, whose positional accuracy is 0.12 mm RMS, is widely used in the medical navigation field [9]. The system is on the basis of BSV. However, BSV has weak anti-occlusion ability. If a camera is blocked, the system stops working. Temporal occlusion greatly increases the risk of using OTS, especially in the medical navigation field, thus, it is remarkable to build a set of strong anti-occlusion, high precision, and low price OTS.

Yachida et al. [10] proposed the use of a third camera information to resolve the ambiguity of binocular matching, after which the multi-camera vision system has been widely developed and used in object reconstruction [11], visual navigation [12], and precision measurement [13]. Baltsavias et al. [14] used the multi-camera stereo vision system to record, model, and visualize the cultural heritage. Conen et al. [15] developed a miniaturized trinocular camera system for 3D reconstruction. Through a series of point cloud reconstruction experiments, they pointed out that the trinocular approach outperformed the binocular approach.

Camera imaging in the NIR band can remarkably reduce the impact of environmental noise on the measurement results and further decrease the complexity of feature extraction [16], thus NIR cameras are widely used for object tracking, such as NDI Polaris [9], Vicon [17], and Qualisys [18]. NIR trinocular stereo vision (TSV) is introduced into OTS in this paper, and main improvements are as follows:A fast nearest neighbor feature point extraction algorithm (FNNF) is proposed. Compared with the traditional pixel-by-pixel search method, the feature points extraction speed is improved by hundreds or even thousands of times, and 30 Hz refresh frame rate can be achieved without the hardware acceleration module.A set of NIR camera calibration devices is designed. Calibration accuracy is improved to 0.02%, and it is superior to the existing calibration methods [19,20,21].The multi-camera 3D reconstruction algorithm based on weights is proposed to further improve the positioning accuracy. Compared with the traditional 3D reconstruction method based on least square method, the accuracy is improved by about 3%.A new active tool is designed. The measurement angle is larger than that of commercial tools.

The rest of this paper is organized as follows: In Section 2, the system setup and principle are described in detail. In Section 3, a series of experiments are conducted to test the feasibility and accuracy of the whole system, then experimental results and discussion are demonstrated in Section 4, followed by the conclusion given in Section 5.

## 2. System Setup and Principal

OTS is a complex tool system composed of hardware and software. Core hardware includes cameras and their accessories as well as optical tools for dynamic tracking. A number of cameras constitute multi-camera stereo vision. Each optical tool has at least three markers. The software is mainly used for the control of cameras, and the tracking and display of tools. Markers are photographed from different angles synchronously. The center of imaging marker corresponds to a projection line in the 3D space, and the intersection point of two or more projection lines is the observed point. Tool tracking is achieved by locating these markers.

### 2.1. Hardware Design

#### 2.1.1. Trinocular Stereo Vision

Optical design is a key consideration for desired imaging results in hardware design, which includes selection of cameras, light-emitting diodes (LEDs), filters, and reflectors. Matching their spectral properties is beneficial to improving the accuracy of OTS. The 850 nm NIR band is selected in this paper to eliminate environmental distractions. Basler acA2000-165 μm NIR complementary metal oxide semiconductor (CMOS) cameras, NIR enhanced, are used for pleasurable image quality in NIR band. The resolution (H × V) is 2048 pixels × 1088 pixels and pixel size (H × V) is 5.5 μm × 5.5 μm. Maximum frame rate is 165 fps. Three Computar M2514-MP2, whose focal length is 25 mm, are utilized as camera lenses. Three FS03-BP850 at the 850 nm NIR band are employed to eliminate ambient light interference.

After TSV calibration, position relationships between cameras need to remain unchanged. Therefore, selection of TSV support is particularly important. The carbon fiber slide track with small thermal expansion coefficient ($7.0\;\times \;10^{-7}/℃$) is adopted to reduce the influence of temperature on the baseline. Baseline length is adjusted by using sliders, and the angle is adjusted by using spherical cradle heads. Two ends are supported by a double tripod. When the system is adjusted to appropriate height, length, and angle, it is locked, as shown in Figure 1. Geometric parameters of OTS are adjusted according to the measurement range.

#### 2.1.2. Active Optical Tool

Real-time location and tracking of optical tools are the ultimate goals of OTS, and the design of optical tools has become an important issue. Tools are mainly divided into active and passive tools [9]. Active tools are generally installed with LEDs, while passive tools are pasted with luminous markers or installed with infrared reflective balls. Both type of tools have some disadvantages. For example, passive tools require an external light source and are susceptible to bad working conditions. If metal surgical instruments or other reflective devices are observed in the working environment, the tracking process may fail. Active tools require additional power supply devices, which increases weight of surgical tools and makes operation inconvenient. Furthermore, due to the uneven imaging of NIR LED, it is rather difficult to obtain high positioning accuracy. However, active tools are more suitable for the surgical environment over passive tools, for extra fake markers caused by other surgical instruments may result in tracking failure. Disadvantages of existing active tools should not be ignored. A new active tool is designed in this paper, which provides outstanding positioning accuracy and a large measurement angle.

Traditional direct-insert LED is used, whose imaging markers from multiple angles are shown in Figure 2a–d. The markers are uneven and vary in shape with different angles, resulting in poor positioning accuracy. To improve uniformity and completeness of imaging markers, a sleeve as shown in Figure 3a is designed. LED is fixed in the sleeve, the inner surface is coated with reflective film to gather light, bottom notches provide space for power supply lines, and a circular hole is opened on the upper for light transmission. To expand the visual angle of the optical marker module, a translucent film is covered on the hole, as shown in Figure 3b. The optical marker module is photographed from different angles, as shown in Figure 2e–h. Markers imaged from large angle (60°) still keep high imaging quality, which is beneficial for high positioning accuracy. Three modules are installed on a tool, and a 1.5 V dry cell is built for the power supply. The active tool model is shown in Figure 3c.

### 2.2. Software Design

Satisfactory hardware design lays a foundation for high precision tracking, and software design is the core of good system. Designs mainly include six parts. Firstly, TSV is controlled for image acquisition. Secondly, NIR camera calibration device is designed for TSV calibration. Thirdly, active tools are photographed by TSV, the captured images are reprocessed, and positions of markers are obtained by using FNNF. Fourthly, 3D reconstruction of markers is performed by using the multi-camera 3D reconstruction algorithm based on weights. Fifthly, the position and posture of the tool are calculated according to the geometric position of markers. The model is processed to be displayed in software. Finally, real-time model display, tracking, and data recording are realized. The system flow chart is shown in Figure 4.

#### 2.2.1. Image Acquisition

Desired images can be captured by setting camera parameters. Basler cameras are used in this system, whose Pylon software development kit (SDK), including drivers, sample program code, and API for C and C++ applications, can be called. It is easy to develop as needs.

acA2000-165 μm NIR cameras support USB 3.0 standard transmission protocol with a maximum transmission rate of 5.0 Gbps, enabling cameras to transmit data at 30 Hz sampling frequency. It should be noted that this system has three cameras. If the upper computer has only one USB 3.0 interface, it is necessary to allocate the bandwidth for each camera, otherwise the phenomenon of frame dropping occurs.

Three cameras are guaranteed to capture images synchronously for OTS. External trigger mode is used in the system and timing of image acquisition is controlled through the same external logic. A microcontroller unit module named ESP32-WROOM-32D is used to generate the 30 Hz trigger signal, and the signal is connected to three cameras. Schematic diagram of the synchronization trigger is shown in Figure 5. The trigger type is configured as a FrameStart mode. The rising edge of the signals is received by the cameras’ optocoupler input interface, then a frame of image is captured and cached in the internal buffer for subsequent processing. Further, 30 Hz frame rate and synchronization are achieved by using 30 Hz square wave.

#### 2.2.2. NIR OTS Calibration

After synchronous camera acquisition has been realized, the calibration of NIR cameras is required to be implemented. To eliminate influences of the ambient light and background on markers extraction, NIR filters are installed in front of cameras. NIR camera calibration has been regarded as a key topic in the field, since NIR cameras cannot capture the texture structures of the visible calibration template. In past decades, many researchers have made explorations in this field. NIR camera calibration methods are generally classified into two types.

Firstly, industrial cameras are capable of imaging both in visible light and NIR wavelengths, thus, cameras can be calibrated in visible light [22]. After calibration, the NIR filter is installed. The disadvantage is that camera parameters may be changed during the process of installing, and effects of wavelength on camera parameters are not considered.

Secondly, a NIR calibration template is designed by using NIR LEDs on a template [23,24]. BSV is used to gain geometric positions of LEDs in visible light, and the geometrical information of the calibration template is obtained. Disadvantages of this approach are obvious. First, the center of LED cannot be accurately measured. Second, LEDs are generally placed in holes of the support template, thus, it is difficult to obtain high mounting precision. Therefore, calibration precision of this method can only reach pixel level, it is far lower than sub-pixel level in visible light.

To solve shortcomings of above calibration methods, a new NIR camera calibration method is proposed in this paper, it combines a Halcon ceramic camera calibration template with a NIR light source at 850 nm band. High precision circular markers with known positions are provided by the template, and illumination is provided by NIR light sources for NIR camera imaging. Relevant literatures state that position extraction accuracy of an ellipse (the result after circular projection) can reach 1/100 pixels [25]. The design of NIR light source is a key factor and following requirements should be met.To reduce chromatic aberration effect on lens, the calibration band of NIR cameras is the same as the working band.Calibration templates are imaged from multiple angles in camera calibration, thus, the brightness of the NIR source is required to be uniform and stable.

The NIR light source has an aluminum template with 100 ${\mathrm{cm}}^{2}$ surface that consists of 100 (10 × 10) NIR (850 nm) LEDs. A high-performance 12 V DC power supply is used to provide stable luminescence for the NIR light source. A lens (108 mm outer diameter, 95 mm inner diameter, 22.5 mm thickness, and 80 mm focal length) is placed 80 mm away from the light source to improve homogeneity by scattering light. To ensure stable and durable working characteristics of the NIR light source, the heat dissipation system is designed. A pure copper template is installed in the back side of LEDs and connected to the rear heat sink by four pure copper heat conduction pipes. The heat dissipation efficiency is improved by installing a fan.

The design of NIR calibration device is completed, the next step is to calibrate NIR TSV. MATLAB Zhang’s calibration package is adopted to calibrate cameras. Halcon circular calibration template is adopted in this design, but the calibration package only can detect checkerboard calibration template, thus, a circular marker extraction algorithm is needed. The gray centroid method based on elliptic boundary is used to extract feature points, the method combines the elliptic fitting with gray centroid method, and it is described in our previous work [26]. After calibration, intrinsic parameters, extrinsic parameters, and distortion coefficients of each camera can be obtained. Both camera parameters and the relative position between three cameras are calculated for TSV calibration [19]. In this paper, the world coordinate system is unified on the calibration template. The first point in the upper left corner of the template is the origin, the horizontal axis is the X axis, the vertical axis is the Y axis, and the vertical calibration template is the Z axis. The calibration diagram of NIR TSV is shown in Figure 6.

#### 2.2.3. Real-Time Image Processing

Multi-directional images of the active tool are synchronously captured by using TSV, then image processing algorithms are performed to detect feature points. Simply searching the whole image for feature points is time-consuming, and it is the bottleneck of real-time processing. Existing processing methods generally use field programmable gate array (FPGA), which has powerful parallel calculating ability. Acquired images are reprocessed by using FPGA, then feature points are transmitted to PC for 3D reconstruction [27]. The calculation burden of PC can be greatly reduced by using this method. However, the FPGA module increases the hardware cost of OTS, and limitations of FPGA are obvious. All functions are realized by hardware, and branch conditional jump cannot be realized. These are not conducive to the secondary development of OTS.

Considering that a small part of the image is occupied by optical markers, if the approximate location can be predicted before conducting a detailed search, and the computation can be greatly reduced, FNNF is proposed. It combines marker prediction, perspective projection, the nearest neighbor fast seed point search algorithm, and region growing algorithm. The flow diagram is shown in Figure 7. Experiments show that the method increases the speed of feature point extraction by hundreds of times over the traditional pixel-by-pixel searching method.

Specific steps of the algorithm are as follows.

Considering the continuity of marker motion and OTS’s real-time tracking, approximate positions of markers in the next frame can be predicted by using simple kinematic equations.After camera calibration, perspective projection equations of each camera are obtained. The position of the predicted point in the image plane can be calculated by perspective projection equations.If the prediction point is on the marker, it is taken as a seed point, and the whole marker is obtained by using the regional growth algorithm. If the predicted point is not on the marker, the seed point search method is used. Considering the region growth algorithm, seed points can be any points on the marker, and markers have a certain size, therefore, searching seed points pixel by pixel is unnecessary. The search starts from the predicted point, search radius, and angle increase from small to large. Search parameters increase at regular intervals according to the actual situation.After whole markers are obtained, the gray centroid method is used to calculate centers.

It should be noted that a certain search radius needs to be set in this method. When the threshold value is reached and seed points are still not found, the marker is treated as occlusion. Working state of OTS can be deduced and adjusted through searching results. For example, when seed points cannot be found in one camera’s image plane, it can be found in the other cameras. This phenomenon means that the camera is occluded. Seed points cannot be searched on the image, which means that the marker is occluded or detached from the field of view or damaged. OTS can be automatically adjusted according to actual situations.

Feature points are extracted by using the traditional method, then the matching relationship of feature points needs to be established for multi-view stereo vision. Common feature point matching methods include epipolar constraint [28,29], ordering and geometrical constraint [30,31]. However, they all have limitations. Epipolar constraint is unable to determine correct correspondences, when two or more markers are coplanar with two optical centers. Ordering and geometrical constraint depend on marker’s location to identify. If parts of markers are blocked, markers will not be identified. In addition, certain restrictions are observed on the number and layout. If FNNF is used, no feature point matching problems are observed. In perspective projection step, corresponding relationships between multiple cameras have been established, thus, no additional matching operations are required. This method not only accelerates extraction speed of feature points, but also solves the problem of mismatching between multiple images.

#### 2.2.4. Stereo Vision

After camera calibration and feature point extraction, 3D reconstruction is carried out. Then, 3D information of spatial objects is obtained from multiple 2D images, which is called stereo vision. As a simple explanation for BSV, it provides four equations (two for each camera) to solve a problem of three unknowns, which is necessary for describing a certain object in space. Due to more spatial information, more accurate 3D reconstruction can be theoretically achieved for multi-camera stereo vision than BSV. In addition, multi-camera stereo vision has redundant positioning ability, thus, the anti-noise ability is greatly enhanced.

Imaging quality and camera parameters (intrinsic parameters, extrinsic parameters, and distortion coefficients) of different cameras are often inconsistent, different cameras may have different effects on reconstruction results in measurement. To further improve accuracy and anti-noise ability, the multi-camera 3D reconstruction algorithm based on weights is proposed. The weight of a camera with large error should be reduced, on the contrary, the weight of a camera with small error should be enhanced. The key of this algorithm is to obtain accurate weights. Detailed derivation of equations can be seen in previously published work [32]. Experiments show that 3D reconstruction accuracy of TSV is higher than that of BSV, and compared with the traditional 3D reconstruction method based on least square method, the accuracy is improved by about 3%.

#### 2.2.5. Tool Processing

After 3D reconstruction, a series of 3D coordinates are obtained. In OTS, multiple active tools are often available, and at least three optical markers are observed on each tool. Therefore, it is particularly important to classify markers. The process of classification is called tool segmentation. Distances are determined after manufacturing, thus, this unique identification code is eligible for tool segmentation in this paper.

The next step is to register tools. The key of registration is to acquire the tip position. The method adopted in this paper is to rotate the tool around a fixed point while ensuring markers be kept within visual measurement range. Coordinates of N positions are collected in total. Since the tool is equivalent to a rigid body, distances between each marker and the tip are regarded as unchanged. The constraint equations are expressed as follows:(1)$${\left(X_{ij}-X_{tp}\right)}^{2}+{\left(Y_{ij}-Y_{tp}\right)}^{2}+{\left(Z_{ij}-Z_{tp}\right)}^{2}=R_{j}{}^{2},$$
where i is ith position, and j is jth marker, $\left(\begin{array}{ccc} X_{ij} & Y_{ij} & Z_{ij} \end{array}\right)$ is the coordinate of ith position, and jth marker in the world coordinate system, $P_{tp}\left(\begin{array}{ccc} X_{tp} & Y_{tp} & Z_{tp} \end{array}\right)$ is the coordinate of the tip in the world coordinate system, $R_{j}$ is the Euclidean distance from the jth marker to the tip.

The current position is subtracted from the initial position, and $R_{j}$ is eliminated. The following equations can be obtained:(2)$$2\left[\begin{array}{ccc} X_{ij}-X_{1j} & Y_{ij}-Y_{1j} & Z_{ij}-Z_{1j} \end{array}\right]\left[\begin{array}{c} X_{tp}\\ Y_{tp}\\ Z_{tp} \end{array}\right]=X_{ij}{}^{2}+Y_{ij}{}^{2}+Z_{ij}{}^{2}-X_{1j}{}^{2}-Y_{1j}{}^{2}-Z_{1j}{}^{2}.$$
For N positions, a total of M markers, M×(n−1) equations can be obtained. The equations are combined:(3)$$GP_{tp}=D.$$
The least squares method is applied to calculate the coordinate $P_{tp}$:(4)$$P_{tp}={\left(G^{T}G\right)}^{-1}G^{T}D.$$Since positional relationships between the tip and each marker are fixed, the tool can serve for a long time after registration. The next step is to calculate the rotation and translation matrix of the tool. As a rigid body, the tool’s rotation and translation matrix can be calculated according to positions of markers before and after. The detailed calculation process is demonstrated in reference [33]. After calculating rotation and translation matrix, position of the needle tip can be calculated directly. The specific calculation equation is shown:(5)$$P_{rtp}=R_{tg}\times P_{tp}+T_{tg},$$
where $P_{rtp}$ is real-time coordinate of the tip in world coordinate system, $R_{tg}$,$\;T_{tg}$ are rotation and translation matrix of the tool, respectively. Tools are tracked in real-time through these operations.

## 3. Experiment

A series of experiments are carried out using self-designed OTS. Experiments mainly include calibration accuracy of NIR camera, positioning accuracy of markers, stability of tool reconstruction, and tracking accuracy of OTS. Figure 8 is the physical map of experimental facilities.

### 3.1. Calibration Precision

Perspective projection relationships between the camera pixel coordinate system and world coordinate system are guaranteed by camera calibration, therefore, camera calibration is the basis of subsequent experiments.

The NIR TSV calibration process is as follows.Cameras are fixed after adjusting their public field of view. The Halcon ceramic camera calibration template is moved in the field of view, and a NIR light source is adjusted to make sure the calibration template image clearly on the image plane. Fifteen images of the calibration template are captured from different angles and positions, respectively. Images captured by NIR TSV are shown in Figure 9.Captured images are input into the Camera Calibration Toolbox for MATLAB to obtain internal parameters, external parameters, and distortion parameters. Partial calibration results are shown in Table 1.The position relationship between cameras is required for multi-camera vision calibration and each camera should be unified to the same world coordinate system. The world coordinate system is unified on the calibration template in first image. External parameters are shown in Table 2.

Where Std represents the uncertainty of optimized results in Table 1. A small Std value indicates high calibration accuracy. The calibration accuracy is relatively high (<0.02%), and comparable to camera calibration accuracy under visible light. To evaluate the advancement of camera calibration methods provided in this study, results of some existing calibration methods are listed in Table 3. The proposed method is better than the other methods.

The measurement range of trinocular vision system needs to be acquired after calibration, as shown in Figure 10. Trinocular overlap area is the effective field of view. The effective field is a maximum of 824 mm wide in the X-axis direction, 437 mm high in the Y-axis direction, and 540–1380 mm in depth direction.

### 3.2. Positioning Accuracy of Luminous Markers

The calculation accuracy of the tool’s rotation and translation matrix are remarkably effected by the positioning accuracy of luminous markers, thus, it is particularly important to evaluate the positioning accuracy. When markers are imaged from different positions and angles, their shape and brightness may change, hence the accuracy needs to be evaluated at different positions and angles.

Distances between two markers are applied to evaluate positioning accuracy. As for the active tool designed in this paper, the distance between marker 1 and 2 (D12) is 54.0634 mm, the distance between marker 2 and 3 (D23) is 59.9596 mm, and the distance between marker 1 and 3 (D13) is 54.0683 mm. The serial number of markers is shown in Figure 11a. To obtain positioning accuracy of markers at different positions and angles, the tool is fixed on the high-precision translation and rotary platform, respectively, as shown in Figure 11b,c.

Positioning experiment at different positions:

A high-precision translation platform is fixed at an appropriate position in the field of view. The tool is fixed on the platform and moved in the direction parallel to the camera baseline (X-axis direction) and depth direction (Z-axis direction), respectively. Each movement distance is 2.00 mm for a single measurement, and a total of 51 position coordinates are obtained for each marker. By connecting coordinates of the same marker, Figure 12a,c can be obtained. The figure shows that motion trajectories of three markers are basically the same, and positioning results are in line with the actual situation. Deviations are calculated to observe the variation intuitively by subtracting distances from the actual value, displayed in the form of scatter diagram in Figure 12b,d. Experimental results indicate that positions have little effect on the positioning accuracy of markers.

To describe the positioning accuracy quantitatively, the standard deviations (STDs) of deviations are calculated. In Figure 12, the red line is the deviation curve of D12, STDs are 0.0070 and 0.0090 mm, respectively. Where the green line is the deviation curve of D23, STDs are 0.0087 and 0.0089 mm, respectively. Where the blue line is the deviation curve of D13, STDs are 0.0087 and 0.0089 mm, respectively. The statistical knowledge states that STD of the distance is $\sqrt{2}$ times as big as that of the single marker [34], so STDs of single marker are 0.0049 and 0.0064 mm, 0.0062 and 0.0063 mm, 0.0035 and 0.0062 mm, respectively. The worst STD is selected as STD of OTS. According to 3σ principle, the positioning accuracy is 0.0192 mm.

Positioning experiment at different angle:

A high-precision rotary platform is fixed at an appropriate position in the field of view. The tool is fixed on the platform, and markers directly face the medium camera of TSV. Angles shown below are all relative to the medium camera image plane. Each movement angle is 2° for a single measurement. When the angle reaches 50°, the tool is rotated in reverse. Fifty-one coordinates are obtained for each marker. By connecting coordinates of the same marker in different pieces, Figure 12e can be obtained. The distance between any two markers is calculated, then it is subtracted from the actual value, and the deviation is plotted as a scatter diagram, as shown in Figure 12f, where the abscissa denotes the angle. The following conclusion can be drawn from the figure: in the field of view, the angle has great influences on the positioning accuracy.

TSV is adopted in this paper. The position and angle of each camera are different, and the influence of the angle is complicated. When the angle is less than 30°, high imaging quality and measurement accuracy of each marker are guaranteed, and the change of angle has little influence on the accuracy. When the angle is greater than 30°, the measurement error increases sharply. The angle of the right camera has already reached 60°, and the deformation of the imaging marker is large with the increase of the angle. When the angle exceeds 40°, the measurement error remains stable. As an explanation, due to poor imaging quality of the right camera, TSV has been reduced to BSV composed of the left and middle camera. When one camera fails, the system can still work normally. The anti-occlusion ability is enhanced. The experimental results also validate that the positioning accuracy of TSV is better than that of BSV. To ensure the high performance of OTS, when a camera fails, it needs to be adjusted in time.

To evaluate positioning accuracy of OTS in the rotation process, the data within 30° is selected as the effective data. STDs are calculated. STD between marker 1 and 2 is 0.0112 mm, STD between marker 2 and 3 is 0.0096 mm, and STD between marker 1 and 3 is 0.0083 mm. STD of single marker is 0.0080, 0.0068, and 0.0059 mm, respectively. The worst reconstruction accuracy is selected as positioning accuracy. According to 3σ principle, the accuracy is 0.0240 mm.

Traditional circular markers are used for visual measurement and medical navigation, and the angle is generally no more than 30° [19,35,36]. The marker designed in this paper can still maintain high positioning accuracy, even the angle reaches 60°. The excellent performance is very beneficial for the application of OTS designed in this paper.

### 3.3. Tool Reconstruction Stability Experiment

Under the influence of light, environment, ground vibration, rounding error in camera acquisition, camera calibration error, and other uncontrollable noise, errors in the positional result of the marker are inevitable. In the case of static tool, reconstruction results are still affected, and the reconstructed tool shakes slightly in visual sense. Additional experiment is conducted to evaluate the stability of the reconstruction.

Active tool registration is required before tool reconstruction. The tool is rotated around a certain point, and multiple positions are photographed. The coordinate of the tip is calculated by Equations (1)–(4).

After the tool registration completed, the coordinate of the tip should be tracked in real-time. The tool is placed in the field of view. Image information is captured synchronously by TSV. Coordinates of three markers are calculated by 3D reconstruction, then coordinates of the tip are calculated by Equation (5). Fifty coordinates are collected in total. The above operation repeats by placing the tool in three different positions within the field of view, and 150 coordinates are collected in total.

After completing the coordinates collection, coordinates should be processed. The mean value of coordinates is taken as the truth value. Euclidean distances between measurements and the truth value are calculated. Figure 13 shows the tracking error under static condition, the abscissa denotes the sequence of image frames. From the figure, we can see that there are little differences between three measurements. The maximum deviation is 0.0374 mm, indicating that the stability of the tool tracking is outstanding. To quantitatively evaluate the stability, STD of 150 data is calculated, and the result is 0.0171 mm. According to the 3σ principle, the static measurement error is 0.0513 mm. The accuracy of a commercial OTS (Polaris Vega, NDI, Canada), which is widely used in many surgical navigation systems, is 0.12 mm RMS [9]. OTS designed in this paper is better than the mainstream OTS.

### 3.4. Evaluation of the Tracking Accuracy

A high precision movement and measurement device is required to evaluate the tracking accuracy. The translation platform is selected, whose translation accuracy is 0.01 mm. The tool is fixed on the platform and moves 2.00 mm for a single measurement. Coordinates of the tool’s tip are recorded, then displacement distances are calculated. The deviation between the measurement distance and the real distance is calculated. A total of 150 data are collected.

After data acquisition, data processing and analysis are carried out. The mean value is −0.0008 mm, which is approximately equals to zero. Conclusions can be drawn, the tracking result is correct, and no or small accumulated error is available. The deviations basically meet the normal distribution, STD is 0.0442 mm, and the mean value is 0.0357 mm. Because the distance is the result influenced by anteroposterior positions, STD of a single position measurement is 0.0313 mm and tracking accuracy is 0.0938 mm. It is better than the tracking accuracy of mainstream OTS. The proposed OTS can be applied for high accuracy dynamic tracking.

## 4. Discussion

A series of experiments are carried out for OTS designed in this paper. In the 3D reconstruction experiment, the tool is placed at different positions, and the positioning accuracy is 0.0192 mm. When the tool rotates in the field of view within ±30°, the positioning accuracy is 0.0240 mm. Comparing the positioning accuracy in rotation and translation experiments, results indicate that the angle still affects the positioning precision within ±30°. Circular optical markers are used in this paper, the center of the imaging markers and target markers does not coincide actually. Due to the existence of center positioning errors, the angle is large, and the positioning accuracy is poor [37]. At present, there are lots of methods to locate the real projection center precisely based on the theory of analytic geometry and spatial perspective [38,39], but how to adapt for the complex working environment and real-time requirements for OTS still needs to be explored.

In the tool tracking experiment, the positioning accuracy of the tip is 0.0513 mm in static measurement, and that is 0.0938 mm in dynamic measurement. They are both better than that measured by existing OTS, but results show that positioning accuracy of the tip is far worse than 3D reconstruction accuracy of single marker. Position of the tip is calculated by locating markers, the relationship between the tip and markers is calculated by registration. The registration method adopted in this paper is to rotate the tool around a fixed point. But it is very difficult to rotate exactly around fixed point and the rotation point is actually not the tip of tool in experiment. These all cause the registration error. The registration method is supposed to be improved in future work.

In addition, the following issues are illustrated concerning the method and results of this article.
(1)Real-time image processing: For the OTS designed in this paper, FNNF is proposed to meet the demand of real-time imaging processing. The computation time of FNNF is closely related to the number and size of the markers. However, the number and size of markers may change with tools moving, thus a certain amount of time redundancy is left in the design of sampling frame rate, and it results in a waste of computing resources.(2)Active surgical tool: The active tool designed in this paper is only used for functional verification. If OTS is used for specific applications, the tool needs to be adapted. For example, in dental implant surgery, active optical markers are installed on the planting tool. The position and posture of the planting tool can be obtained by tracking optical markers. The tool is used to drill holes in the patient’s missing teeth position by doctors. Both the depth and angle of the hole are monitored by the designed OTS to minimize the surgical trauma. This system is able to increase the success ratio of surgery and lower the surgery risks.(3)Occlusion: The anti-occlusion ability of OTS is improved by using NIR TSV and new active tools, but the occlusion problem cannot be completely solved. For example, when a marker is completely occluded, each camera cannot detect it. The exploration of the occlusion problem is the focus of the following work.

## 5. Conclusions

OTS based on NIR TSV is built and described in this paper, including all of its components and the structure of these components, as well as components of software system.

To eliminate the influence of ambient light on the tracking process, this system works in the NIR band. The NIR active tool is designed to overcome the shortcomings of existing tools in this paper, providing outstanding positioning accuracy and a large measurement angle (60°). Due to the poor precision of existing NIR calibration methods, a new calibration method is proposed by combining the NIR light source and Halcon ceramic calibration template. Calibration accuracy is improved to 0.02%, which is higher than that of existing methods. Compared with BSV, more image information is provided by TSV, and higher accuracy of 3D reconstruction can be achieved, but heavy processing burden is caused by a large amount of redundant image information. To solve the problem, FNNF is presented. It combines location prediction with nearest neighbor feature points search. The 30 Hz refresh frame rate can be achieved without extra hardware accelerate units. Then, 3D reconstruction of feature points is conducted after feature point extraction. Considering different cameras with different measurement errors, contributions of different cameras on 3D reconstruction accuracy should be different. The multi-camera 3D reconstruction method based on weights is proposed, and the accuracy is improved by about 3%. Finally, the tool is processed to achieve real-time tracking and display. In the tool tracking experiment, the dynamic tracking accuracy can reach 0.0938 mm.

Compared with the existing OTS, the designed OTS has higher precision, stronger anti-occlusion, and lower cost owing to innovations of the hardware and software, and it is beneficial for medical navigation.

## Figures and Tables

**Figure 1 sensors-21-02528-f001:**
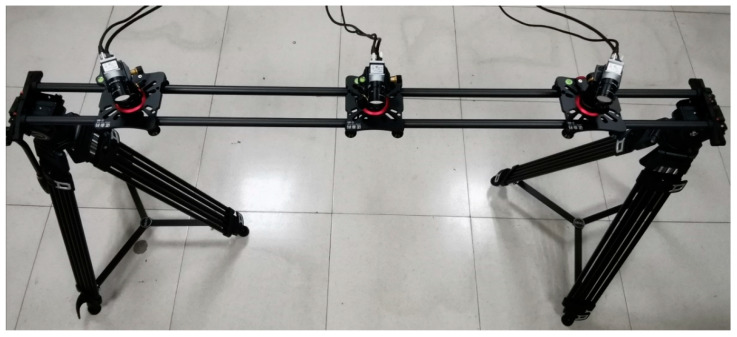
Physical map of trinocular stereo vision (TSV).

**Figure 2 sensors-21-02528-f002:**
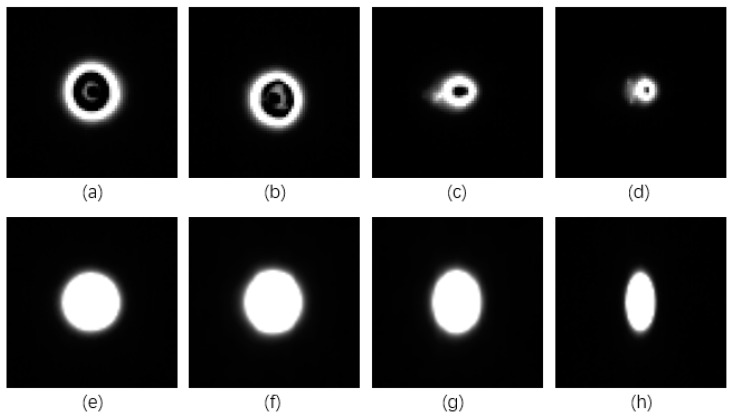
Near infrared (NIR) markers imaged from different angles. (**a**) Traditional marker imaging result from 0°. (**b**) Traditional marker imaging result from 20°. (**c**) Traditional marker imaging result from 40°. (**d**) Traditional marker imaging result from 60°. (**e**) Designed marker imaging result from 0°. (**f**) Designed marker imaging result from 20°. (**g**) Designed marker imaging result from 40°. (**h**) Designed marker imaging result from 60°.

**Figure 3 sensors-21-02528-f003:**
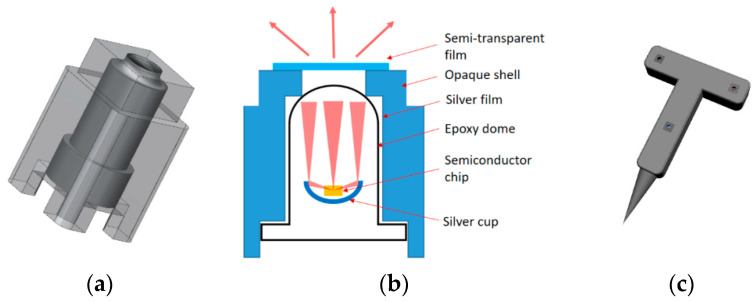
Schematic diagram of an active tool. (**a**) Model diagram of light-emitting diodes (LED) sleeve. (**b**) Active marker schematic. (**c**) Active tool model.

**Figure 4 sensors-21-02528-f004:**
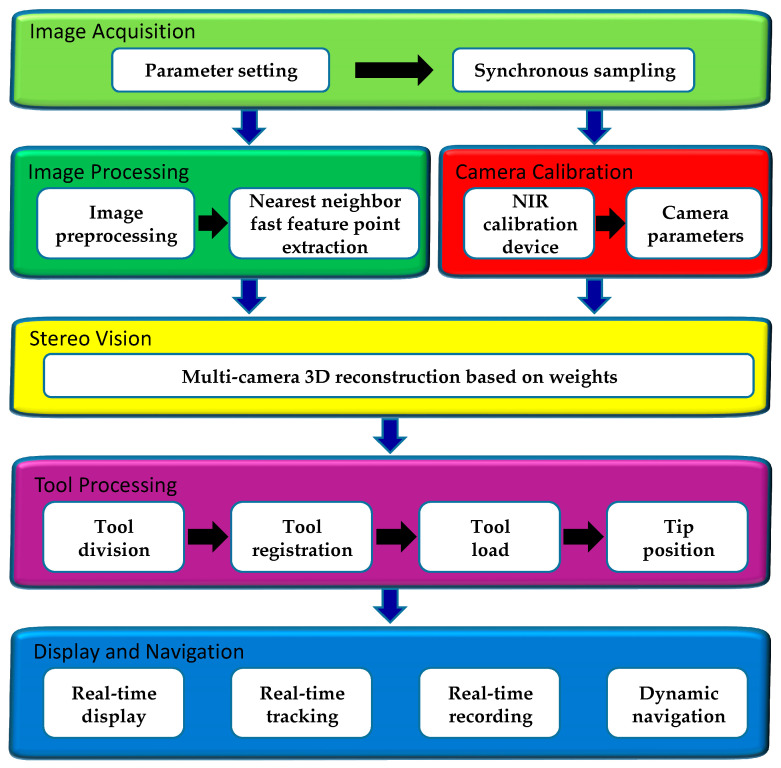
The flow chart of the software.

**Figure 5 sensors-21-02528-f005:**
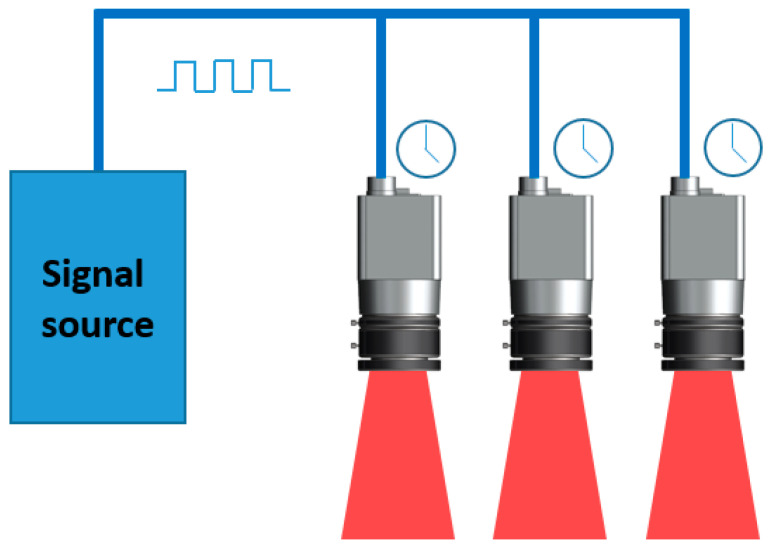
Schematic diagram of synchronous acquisition of TSV.

**Figure 6 sensors-21-02528-f006:**
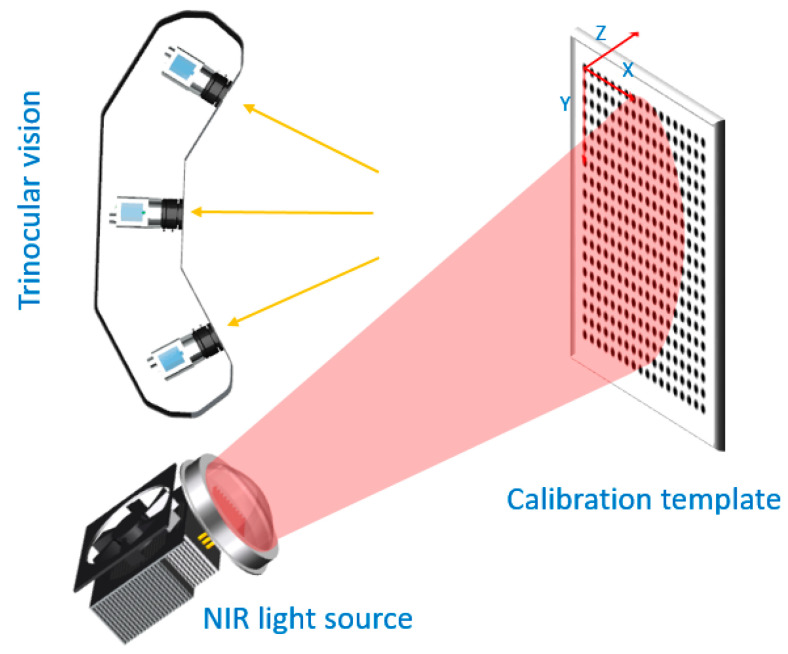
NIR trinocular stereo vision calibration.

**Figure 7 sensors-21-02528-f007:**
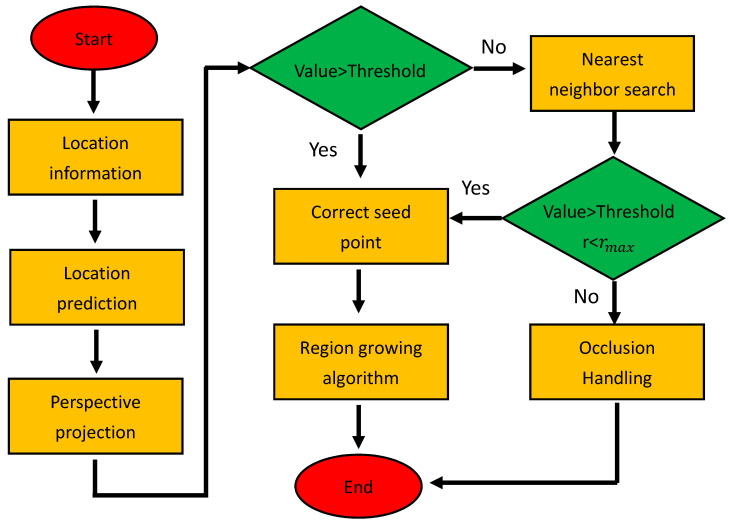
The flow diagram of fast nearest neighbor feature point extraction algorithm (FNNF).

**Figure 8 sensors-21-02528-f008:**
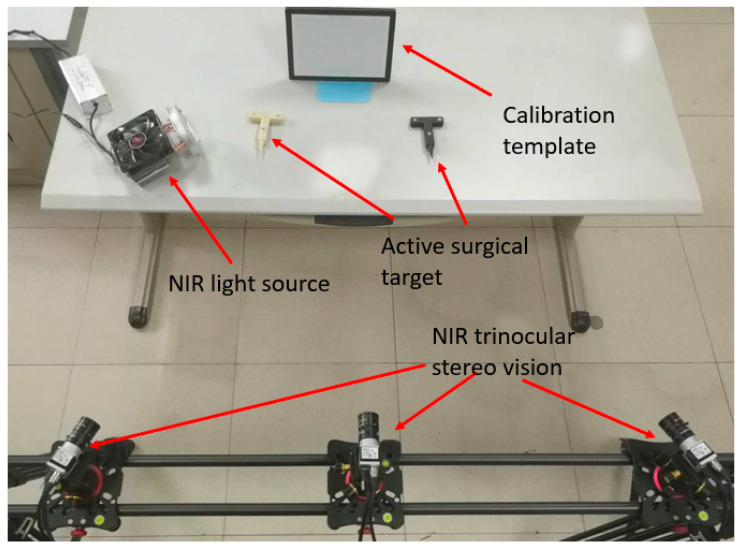
System setup.

**Figure 9 sensors-21-02528-f009:**
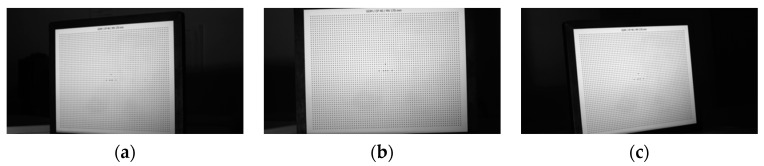
The calibrate template captured by three cameras. (**a**) Imaged by the left camera. (**b**) Imaged by the medium camera. (**c**) Imaged by the right camera.

**Figure 10 sensors-21-02528-f010:**
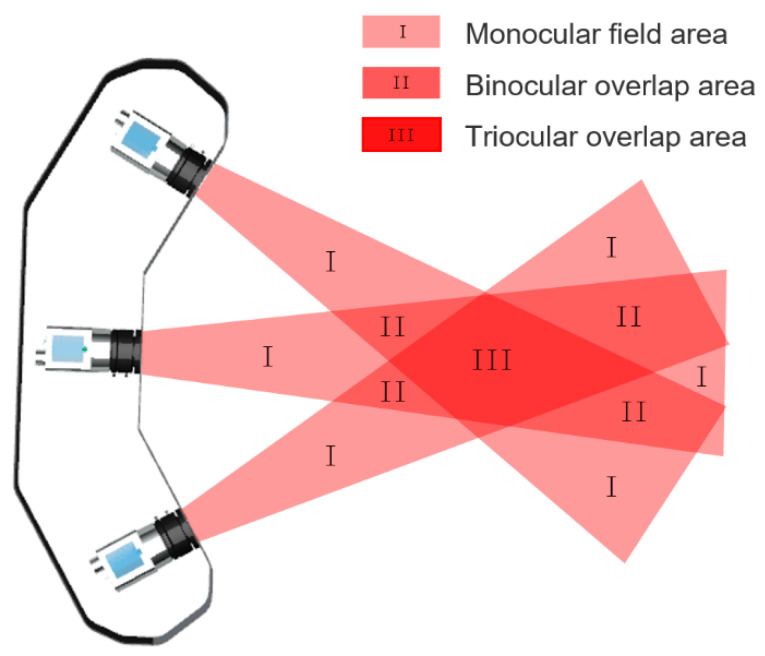
Measurement range of the system.

**Figure 11 sensors-21-02528-f011:**
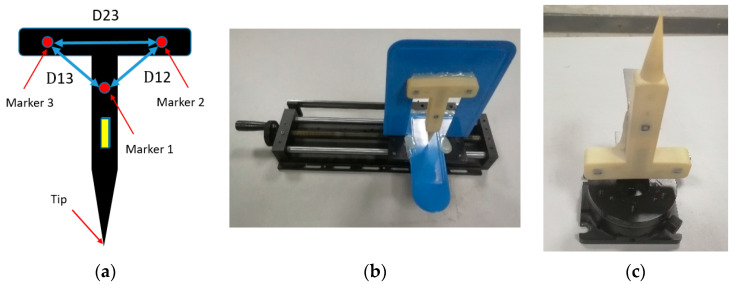
Experimental facilities. (**a**) Model diagram. (**b**) A high-precision translation platform fixed with a tool. (**c**) A high-precision rotary platform fixed with a tool.

**Figure 12 sensors-21-02528-f012:**
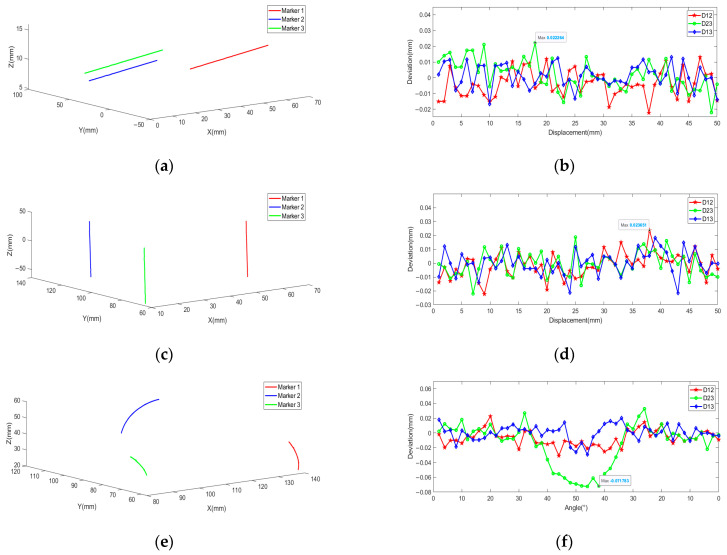
Track and deviation charts at different positions and angles. (**a**) Track of movement along the X axis. (**b**) Deviation of movement along the X axis. (**c**) Track of movement deviation along the Z axis. (**d**) Deviation of movement along the Z axis. (**e**) Track of rotation along the Y axis. (**f**) Deviation rotation along the Y axis.

**Figure 13 sensors-21-02528-f013:**
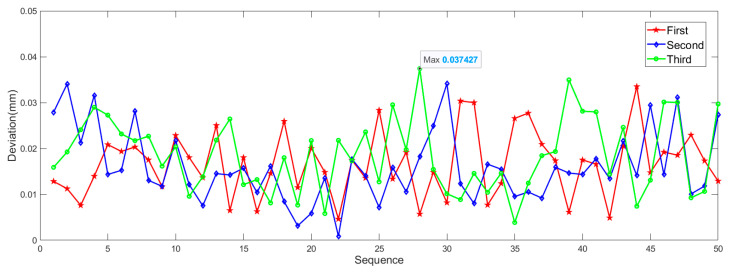
Tracking error under static condition.

**Table 1 sensors-21-02528-t001:** Camera calibration parameters and related errors. Std represents the uncertainty of the optimized results.

		Left		Medium		Right	
		Value	Std	Value	Std	Value	Std
Focal length (pixels)	X	4807.032	0.719	4812.444	0.883	4787.796	0.709
	Y	4807.235	0.667	4812.175	0.873	4787.319	0.706
Principal point (pixels)	U	1032.466	0.808	1017.746	0.885	1018.847	0.989
	V	548.330	0.638	546.137	0.679	556.095	0.741

**Table 2 sensors-21-02528-t002:** Camera calibration exterior parameters.

	Left Camera	Medium Camera	Right Camera
Rotation matrix	$\left[\begin{array}{ccc} 0.0120 & 0.8585 & -0.5127\\ 0.9997 & -0.0206 & -0.0112\\ -0.0202 & -0.5124 & -0.8585 \end{array}\right]$	$\left[\begin{array}{ccc} 0.0257 & 0.9996 & -0.0107\\ 0.9997 & -0.0257 & -0.0059\\ -0.0062 & -0.0105 & -0.9999 \end{array}\right]$	$\left[\begin{array}{ccc} 0.0902 & 0.8516 & -0.5164\\ 0.9958 & -0.0706 & -0.0576\\ -0.0126 & 0.5194 & -0.8544 \end{array}\right]$
Translation matrix(mm)	$\left[\begin{array}{c} -11.1325\\ -59.8417\\ 948.9626 \end{array}\right]$	$\left[\begin{array}{c} -96.3602\\ -73.9670\\ 779.6404 \end{array}\right]$	$\left[\begin{array}{c} -92.1928\\ -57.1111\\ 886.2988 \end{array}\right]$

**Table 3 sensors-21-02528-t003:** Comparison of results obtained from different calibration methods (unit: pixels). “Proposed” represents the method proposed in this paper.

Method	X_Value	X_Std	Y_Value	Y_Std
Yang et al. [19]	2055.880	4.940	20.540	5.080
Zheng et al. [20]	2509.774	5.790	2509.756	6.049
Vidas et al. [21]	638.850	1.350	655.240	1.330
Proposed	4812.444	0.883	4812.175	0.873

## Data Availability

Not applicable.

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
