# Peer review of "High Precision Optical Tracking System Based on near Infrared Trinocular Stereo Vision"

_sensors, 2021, doi:10.3390/s21072528_

Round 1

Reviewer 1 Report

This manuscript presents a high precision optical tracking system (OTS) based on near infrared trinocular stereo vision. This system has advantages with respect to traditional OTS based on binocular stereo vision. However, this version of manuscript must be improved considering the following comments:

1.-English style and grammar of all the sections in the manuscript must be significantly improved.

2.-In second section, the writing of the following sentences should be improved:

What’s more, due to the non-uniformity of 110 NIR LED imaging, improvements of positioning accuracy are limited.

The markers are uneven and have great differences in different 117 angles with poor positioning accuracy.

The optical marker module is placed at different angles for imaging, and images are 123 shown in Fig. 2(e-h). Markers imaged from large angle (60°) still maintain completeness 124 with high positioning accuracy

3.-The following sentence should have a reference.

4.-The quality of the figures 4, 7, 12, and 13 must be enhanced.

5.-Table 1 should be revised.

6.-Table 2 has different size of the numbers.

7.-Authors should add more discussion about the results of 12 and 13.

8.-Authors should include the main limitations or challenges of the proposed (OTS) based on near infrared trinocular stereo vision.

9.-References have format of IEEE. References must be writen using the format of Sensors.

Reviewer 2 Report

The paper “High precision optical tracking system based on near infrared trinocular stereo vision” proposes a device based on three cameras arranged to ensure a stereo vision of the target object in the near infrared, to improve the spatial precision of detection of target issues. The paper is well written, experimental set up, data processing and validation are carefully described and convincing.  Considering the still experimental arrangement of the device, the only remark is the requirement of a more specific statement of conceivable applications, i.e. the surgery fields mostly likely getting benefice from the assistance from this device.

Reviewer 3 Report

In general, this paper has sufficient standard and is well structured to be accepted for publication. It involves interesting results and good elaboration on both theoretical and practical research. However, the contribution of this work is not clearly highlighted and some parts of the paper needs more explanation. I assume the 4 items at the end of Section one are the contribution but not explicitly stated. What is the novelty in this work?

It is recommended to expand the application of OTS and the proposed system. For example, references [1-8] was solely cited and not reviewed.

It is not clear to me how the system works particularly the hardware part. For instance, in 2.1.2. “tracking position” what you are referring to?

When referring to Fig. 4, it is beneficial to describe various steps.

The figure at line 162 does not have any caption.

Section 2.2.3 should be “real-time image processing”

Round 2

Reviewer 1 Report

This version of manuscript has been significantly improved considering the reviewer's comments.